# Anomalous frozen evanescent phonons

Yi Chen [1,2,5] ✉, Jonathan L. G. Schneider [2,5], Ke Wang[2,3,5], Philip Scott[2], Sebastian Kalt [2], Muamer Kadic [4] & Martin Wegener [1,2] ✉

Evanescent Bloch waves are eigensolutions of spatially periodic problems for complex-valued wavenumbers at finite frequencies, corresponding to solutions that oscillate in time and space and that exponentially decay in space. Such evanescent waves are ubiquitous in optics, plasmonics, elasticity, and acoustics. In the limit of zero frequency, the wave "freezes" in time. We introduce frozen evanescent waves as the eigensolutions of the Bloch periodic problem at zero eigenfrequency. Elastic waves, i.e., phonons, in metamaterials serve as an example. We show that, in the complex plane, the Cauchy-Riemann equations for analytical functions connect the minima of the phonon band structure to frozen evanescent phonons. Their exponential decay length becomes unusually large if a minimum in the band structure tends to zero and thereby approaches a soft mode. This connection between unusual static and dynamic behaviors allows to engineer large characteristic decay lengths in static elasticity. For finite-size samples, the static solutions for given boundary conditions are linear combinations of frozen evanescent phonons, leading to interference effects. Theory and experiment are in excellent agreement. Anomalous behavior includes the violation of Saint Venant's principle, which means that large decay-length frozen evanescent phonons can potentially be applied in terms of remote mechanical sensing.

Bloch waves in natural as well as artificial periodic materials can be described by their dispersion relation, i.e., by the dependence of the wave angular frequency $\omega_i(k)$ on the wavenumber $k$ (or, more generally, on the wavevector)[1–4]. The integer subscript $i = 1, 2, \ldots$ is the band index. In the vast majority of experimental situations, one considers exciting the material in some form at a pre-described real-valued frequency $\omega$. Within the linear response, the driven system reacts with real-valued frequency $\omega_i = \omega$. For propagating waves, by definition, the wavenumber $k$ is always real-valued as well[5]. However, for evanescent modes, i.e., for modes exponentially decaying (or increasing) in amplitude along one or more directions, the wavenumber $k$ is complex-valued. Its real part determines the wavelength $\lambda = 2\pi/\mathrm{Re}(k)$, and its imaginary part determines the exponential decay length $l = 1/\mathrm{Im}(k)$.

Ordinarily, evanescent waves are associated with finite frequencies inside frequency bandgaps[6]. A well-known example is the Jackiw-Rebbi solution[7–9] localized at a domain wall separating two different topological phases. Jackiw-Rebbi states have been applied to study static domain wall states and corner modes in mechanical metamaterials[10]. Here, we apply the concept of evanescent waves in single-domain samples to the *static* regime, in which the wave gets "frozen" in time. These special Bloch modes allow for making a direct connection between unusual static and dynamic properties, which has previously been unclear, hampering the rational design of unusual static behavior in metamaterials. The Bloch eigenmodes of elastic waves in metamaterials, phonons, serve as an example. By a general discussion based on the Cauchy-Riemann equations, treating the band structure as an analytical function, we introduce the concept of frozen

[1]Institute of Nanotechnology, Karlsruhe Institute of Technology (KIT), Karlsruhe, Germany. [2]Institute of Applied Physics, Karlsruhe Institute of Technology (KIT), Karlsruhe, Germany. [3]National Key Laboratory of Science and Technology on Advanced Composites in Special Environments, Harbin Institute of Technology, Harbin, China. [4]Université de Franche-Comté, Institut FEMTO-ST, UMR 6174, CNRS, Besançon, France. [5]These authors contributed equally: Yi Chen, Jonathan L. G. Schneider, Ke Wang. ✉e-mail: yi.chen@partner.kit.edu; martin.wegener@kit.edu

evanescent phonons. We show analytically that the frozen-phonon decay length may diverge as a minimum in the band structure approaching zero frequency at some wavenumber $k = k_{min}$ in the Brillouin zone. Taking nonlocal mechanical metamaterials[11–13] as a class of examples, we compare numerical simulations and calculations based on a simplified mass-and-spring model with experiments on finite microstructured three-dimensional (3D) metamaterial beams. The resulting unusual static behavior shows commensurability effects and interference effects and violates Saint Venant's principle[14,15]. The latter means that the reaction of the system far away from a point at which a force is applied changes substantially even if one shifts this point by only a single unit cell. Furthermore, we discuss the implications of anomalous frozen evanescent phonons for the lowest-frequency eigenmodes of finite-size beams or plates. We find that even the fundamental eigenmode of a guitar string or a drum membrane made of an anomalous material can exhibit pronounced spatial oscillations – in sharp contrast to ordinary behavior.

## Results

### Frozen evanescent phonons

Let us start our discussion from general principles, independent from any particular experimental realization: The phonon band structure $\omega_i(k)$ results from a physical equation of motion of the system and is thus an analytical function, at least in the vicinity of the real axis in the complex plane[16,17]. Consider a minimum or maximum of the real frequency $\omega_i$ in the band structure of the band $i$ at the real wavenumber $k_{min/max}$ illustrated in Fig. 1. In the vicinity of such an extremum, we can Taylor expand

$$\omega_i(\mathrm{Re}(k)) = \omega_{min/max} + \frac{1}{2}\zeta\Big(\mathrm{Re}(k) - k_{min/max}\Big)^2 + \ldots \quad (1)$$

Mathematically, frequency $\omega_i$ and wavenumber $k = \mathrm{Re}(k) + i\,\mathrm{Im}(k)$ can be complex numbers. Thus, the function $\omega_i(k) = f(z) = u(x + iy) + i\,v(x + iy)$ follows the Cauchy-Riemann equations[18]

$$\frac{\partial u}{\partial x} = \frac{\partial v}{\partial y}; \frac{\partial u}{\partial y} = -\frac{\partial v}{\partial x} \Rightarrow \frac{\partial^2 u}{\partial x^2} = -\frac{\partial^2 u}{\partial y^2}, \quad (2)$$

hence

$$\frac{\partial^2(\mathrm{Re}(\omega_i))}{\partial(\mathrm{Re}(k))^2}(k_{min/max}) = -\frac{\partial^2(\mathrm{Re}(\omega_i))}{\partial(\mathrm{Im}(k))^2}(k_{min/max}) = \zeta. \quad (3)$$

Therefore, as illustrated in Fig. 1, a minimum with $\zeta > 0$ (maximum with $\zeta < 0$) leads to branches of evanescent modes toward lower (higher) frequencies than the extremum. We are mainly interested in the modes toward lower frequencies, which can arrive at zero frequency. If the minimum in the band structure approaches zero, $\omega_{min} \to 0$, we can again Taylor expand, truncate, and compute the point at which the evanescent branch hits the $\omega = 0$ plane in Fig. 1. In this plane, the phonon wave gets frozen. A finite $\mathrm{Re}(k_{min})$ means that the frozen-phonon oscillation has a spatial period (to avoid the notion "wavelength") of $p = 2\pi/\mathrm{Re}(k_{min})$. The corresponding imaginary part, $\mathrm{Im}(k_{min})$, means that the frozen-phonon oscillation decays exponentially with exponential decay length $l$ given by

$$l = \frac{1}{|\mathrm{Im}(k_{min})|} = \sqrt{\frac{\zeta}{2\omega_{min}}}. \quad (4)$$

Clearly, this decay length diverges, $l \to \infty$, as the frequency at the minimum (with $\zeta > 0$) approaches zero, $\omega_{min} \to 0$. Thus, this relation connects unusual static and unusual dynamic phonon behavior in materials and metamaterials. In terms of design, we take advantage of the fact that minima in the dispersion relation with $\omega_{min} \to 0$ have recently been designed rationally along different lines[11,19–21].

For finite-size materials subject to static boundary conditions imposed from the outside, the solution of the frozen-phonon displacement field is given by that superposition of the discussed evanescent frozen-phonon eigensolutions that obey these boundary conditions. In Fig. 1, for clarity, we depict only the two eigensolutions with $\mathrm{Re}(k) > 0$. Assuming reciprocity (which is given for a linear passive lossless problem obeying time-inversion symmetry), two further eigensolutions with $\mathrm{Re}(k) < 0$ arise from flipping the sign, i.e., from $\omega_i(\mathrm{Re}(k),\mathrm{Im}(k)) = \omega_i(\pm\mathrm{Re}(k),\pm\mathrm{Im}(k))$. As a result, the actual solution for the *finite* system depends not only on the discussed eigensolutions for the *infinite* periodic system but also on the boundary conditions and, hence, on the size of the specimen. For large decay length $l$ this means that, e.g., moving the point at which a force is applied by one

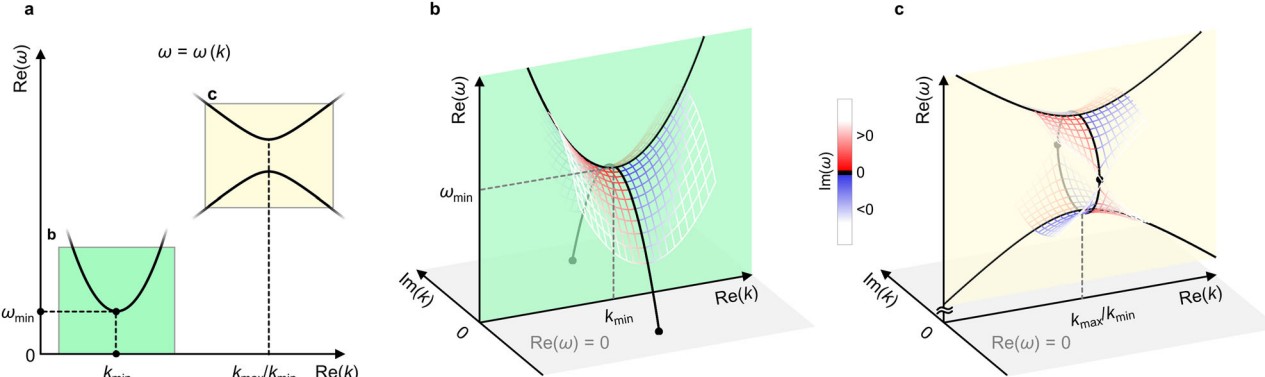

**Fig. 1 | Complex-valued band structures and frozen evanescent Bloch modes.** A linear, passive, and lossless infinite periodic system is considered. **a** Two possible scenarios of dispersion relations, exhibiting local extrema of the real part of the eigenfrequency, $\mathrm{Re}(\omega) > 0$, versus the real part of the wavenumber, $\mathrm{Re}(k) > 0$. The green and yellow colors refer to panels b and c, respectively. **b** Real part of the eigenfrequency versus real and imaginary part of the wavenumber for the scenario highlighted in green in (**a**). The black curves correspond to $\mathrm{Im}(\omega) = 0$. The imaginary part of $\omega$ is false-color coded. Two evanescent branches with $\mathrm{Im}(k) \neq 0$ emerge from the local minimum of $\mathrm{Re}(\omega)$ versus $\mathrm{Re}(k)$ (green plane) and touch the $\omega = 0$ plane (gray plane). These static or frozen eigenmodes are highlighted by the two black dots. For $\mathrm{Re}(k) < 0$, two further such modes occur. The characteristic exponential decay length of these frozen modes is given by $l = 1/|\mathrm{Im}(k)|$, their spatial oscillation period by $p = 2\pi/\mathrm{Re}(k)$. **c** Same as panel b, but for the scenario highlighted in yellow in panel a. Here, no evanescent zero-frequency Bloch modes result.

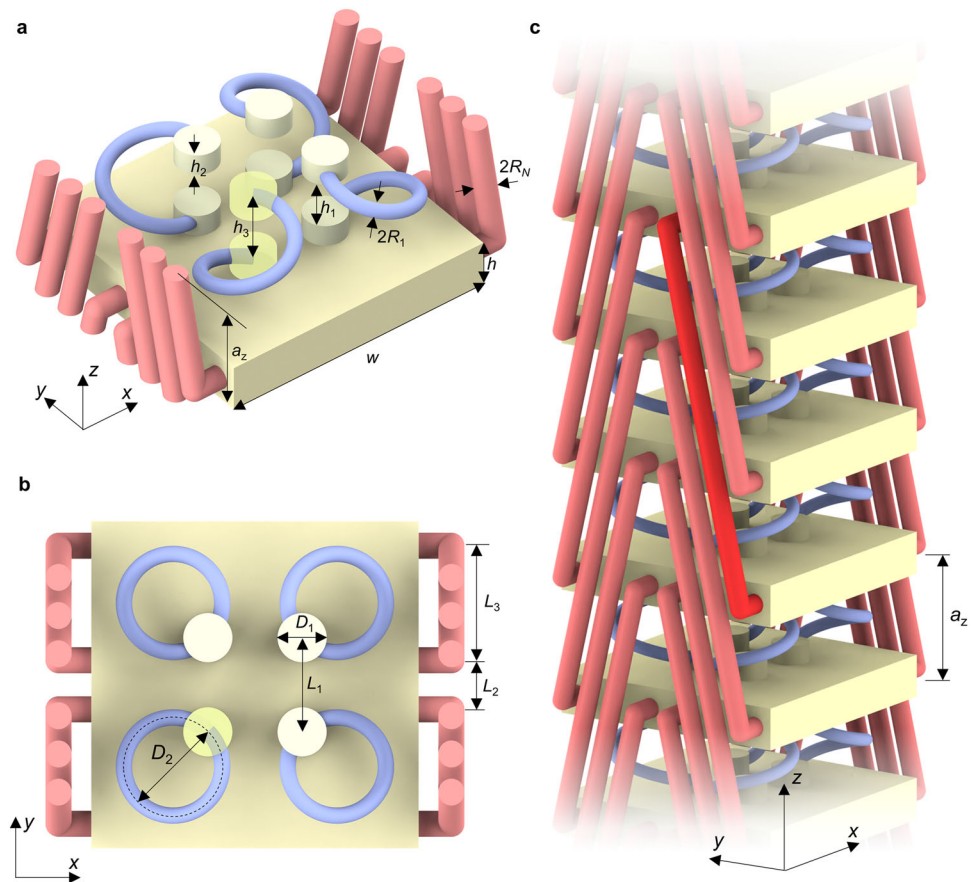

**Fig. 2 | Blueprint of metamaterial supporting anomalous frozen evanescent phonons.** This nonlocal mechanical metamaterial composed of a single constituent polymer material allows obtaining frozen evanescent Bloch modes with large characteristic exponential decay length $l$ (cf. Fig. 1b). **a, b** Two different views onto a single unit cell. The geometrical parameters are defined. Two of the yellow cylinders are rendered semi-transparent to indicate the height of blue helices. The colors are for illustration only. **c** Beam composed of a one-dimensional periodic arrangement of this unit cell along the $z$-direction with period $a_z$. Adjacent yellow plates are connected by four blue helices ("springs"), fixed to the plates by yellow cylinders. The handedness of the springs alternates, such that the overall structure has two mirror planes, making it achiral. The strength of this nearest-neighbor interaction can be tailored by the radius $R_1$. The yellow plates are additionally connected to their $N$-th neighbors by the red rods with radius $R_N$. This radius determines the strength of the $N$-th nearest neighbor interactions. The example shown refers to $N = 3$. We will discuss $N = 2, 3, 4$ with different geometrical parameters. The geometric parameters for $N = 2, 3$, and 4 are chosen as $2R_1/a_z = 0.10$, $2R_N/a_z = 0.156$, $2R_1/a_z = 0.10$, $2R_N/a_z = 0.16$, and $2R_1/a_z = 0.072$, $2R_N/a_z = 0.10$, respectively. All other geometrical parameters are fixed: $a_z = 100\,\mu m$, $2R_1/a_z = 0.10$, $2R_N/a_z = 0.16$, $w/a_z = 2.0$, $h/a_z = 0.34$, $h_1/a_z = 0.16$, $h_2/a_z = 0.34$, $h_3/a_z = 0.50$, $D_1/a_z = 0.30$, $D_2/a_z = 0.60$, $L_1/a_z = 0.57$, $L_2/a_z = 0.30$, and $L_3/a_z = 0.70$. For the material parameters of the constituent polymer, we choose mass density $\rho = 1190\,kg/m^3$, Young's modulus $E = 4.19$ GPa, and Poisson's ratio $v = 0.3$.

unit cell may substantially change the response of the material far away from that point.

In the following example, we illustrate the concept and the implications of frozen evanescent waves for phonons in nonlocal metamaterials. For this example, the minima of $\omega_i(\mathrm{Re}(k))$ for $i = 1$ occur at wavenumbers inside of the first Brillouin zone, $0 < |\mathrm{Re}(k)| < \pi/a$, leading to frozen real-space oscillations with $l \gg a$ and period $p \approx N\,a$, where $N$ is the integer order of nonlocal interactions that occur in addition to the nearest-neighbor interactions. At the end of this paper, we briefly address other examples.

## Complex band structure and Bloch eigenmodes

Following our introduction, to achieve anomalous frozen evanescent Bloch modes with decay lengths much larger than one metamaterial unit cell, we can start from any mechanism leading to a local minimum of the dispersion relation $\omega_1(\mathrm{Re}(k)) \to 0$ that approaches zero frequency. Several approaches to achieve such local minima have been published[22–25]. Here, we choose nonlocal metamaterials with strong beyond-nearest-neighbor interactions[26–32]. This approach has proven to be particularly flexible[33–36]. In the limit that the beyond-nearest-neighbor interactions are much stronger than the nearest-neighbor

interactions, a local minimum in the acoustical phonon dispersion relation approaches zero frequency[11]. The 3D mechanical metamaterial suggested in Fig. 2 allows us to conceptually tailor the relative strengths of local and nonlocal interactions over a large range, and in a manner that is amenable to state-of-the-art 3D manufacturing. Figure 2a, b depicts and defines the unit cell. However, from inspecting only a single unit cell, the role of the nonlocal interactions is difficult to comprehend. Figure 2c shows a metamaterial beam that results from repeating the unit cell along the $z$-direction with period $a_z$. The four light-blue helical rods or "springs" with diameter $2R_1$ are responsible for connecting the plates in two neighboring unit cells and thereby constitute the local interactions. We note that these springs intentionally have different handedness and are arranged in a manner that leads to mirror symmetry with respect to the $xz$- and the $yz$-planes. The metamaterial is thus achiral, avoiding a twist behavior that can be interesting in general[37–40], but that would unnecessarily complicate the behavior here. The yellow cylinders merely fix the springs to the plates in a well-defined manner. The red cylindrical rods with diameter $2R_N$ serve to couple the plates to their $N$-th neighbors. Thereby they constitute the nonlocal interactions. In Fig. 2, $N = 3$ is used as an example. In our below calculations and experiments, we will consider $N = 2$,

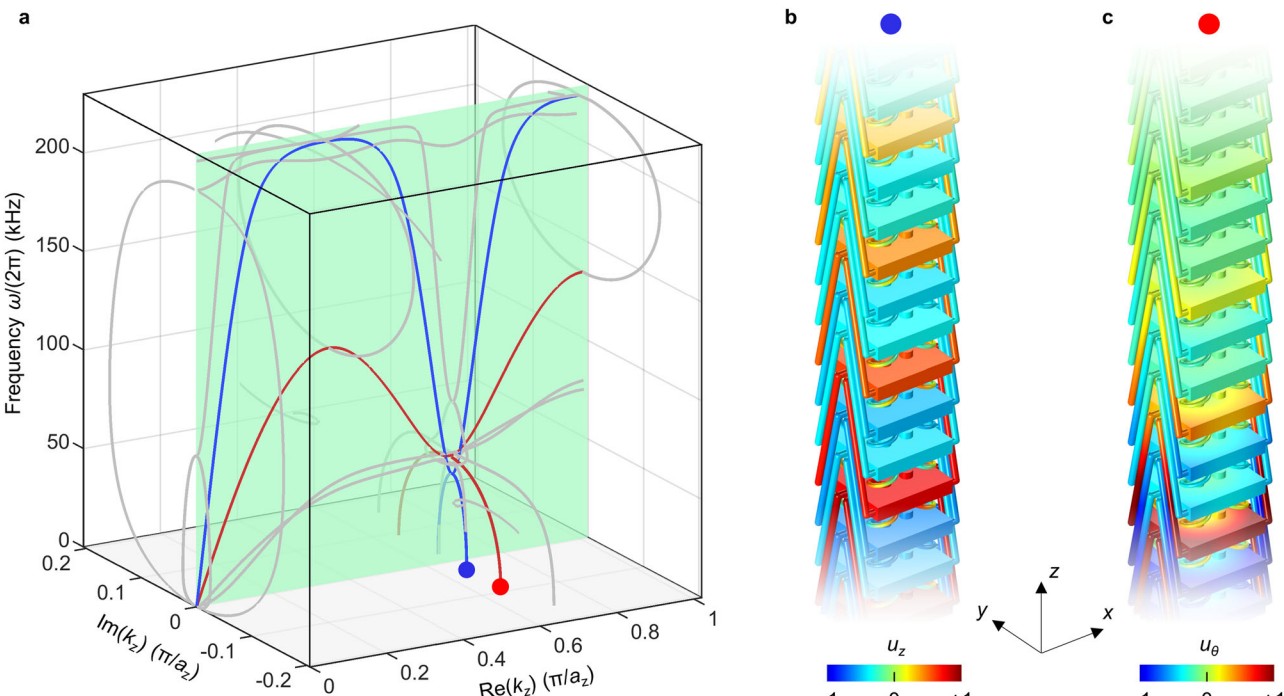

**Fig. 3 | Complex-valued phonon band structure and frozen evanescent phonon modes. a** Numerically calculated phonon band structure for elastic-wave propagation in the metamaterial beam ($N=3$) defined in Fig. 2 for wave propagation along the $z$-direction. The representation is as in Fig. 1b, except that only real-valued frequencies are depicted, $\text{Im}(\omega) = 0$. Out of many modes (gray), two are highlighted. The blue modes correspond to longitudinal waves, and the red modes to twist waves. Out of the corresponding local minima in the green plane, evanescent modes emerge (cf. Fig. 1b) that touch the $\omega = 0$ plane at the positions of the colored dots. For the longitudinal mode (blue dot) relevant to the below experiments, we find the complex-valued wavenumber $k_z \approx (0.666 - 0.026\,\text{i})\,\pi/a_z$. Similar band structures are shown in Supplementary Fig. 1 for $N=2$ and $N=4$. **b** Illustration of this frozen mode. The axial component of the displacement vector, $u_z$, is depicted in a false-color representation. The static spatial oscillation period of $p = 2\pi/\text{Re}(k_z) \approx Na_z = 3a_z$ is clearly visible. The mode exponentially decays with decay length $l = 1/|\text{Im}(k_z)|$. **c** Same as panel (**b**), but for the zero-frequency twist mode (red dot in **a**), with an azimuthal component of the displacement vector $u_\theta$.

$N = 3$, and $N = 4$. The different colors in Fig. 2 are for illustration only. All parts actually correspond to the same constituent material. In our experiments based on 3D laser microprinting, we use a polymer, which is well described by the following static elastic parameters: mass density $\rho = 1190\,\text{kg/m}^3$, Young's modulus $E = 4.19\,\text{GPa}$, and Poisson's ratio $\nu = 0.3$.

It is straightforward to map the metamaterial beam shown in Fig. 2 onto a simple one-dimensional mass-and-spring model[11] containing local Hooke's springs with spring constant $K_1$ and nonlocal Hooke's springs with spring constant $K_N$ (for parameters, see "Methods"). Below, we will compare the results of this simple approximation with both finite-element continuum-mechanics calculations and experiments.

In Fig. 3a, we show the complex-valued phonon band structure $\omega_i(k_z)$ of the metamaterial beam in Fig. 2c, numerically calculated using linear continuum mechanics (see "Methods"). The blue (red) curve in the light-green plane refers to the usual longitudinal (twist) propagating phonon Bloch mode with real phonon wavenumbers (for clarity, other bands corresponding to flexural and bending modes are plotted in gray). As discussed previously[11], due to the strong third-order nonlocal interactions ($N = 3$), both bands exhibit pronounced local minima at a wavenumber around $k_z = 2\pi/(3a_z)$. Evanescent modes with non-zero $\text{Im}(k_z)$ emerge from these minima and reach down to zero frequency, $\omega = 0$, leading to frozen evanescent Bloch modes. For the longitudinal frozen mode (blue dot), we have $k_z \approx (0.666 - 0.026\,\text{i})\,\pi/a_z$, leading a characteristic decay length of $l = 1/|\text{Im}(k_z)| \approx 12.2\,a_z$. For the twist mode (red dot), we obtain $k_z \approx (0.659 - 0.093\,\text{i})\,\pi/a_z$, thus $l = 1/|\text{Im}(k_z)| \approx 3.4\,a_z$. If the relative strength of the nonlocal interactions is further increased,

e.g., by decreasing $2R_1$ (cf. Fig. 2) while fixing all other parameters, both minima decrease in frequency, and the decay lengths further increase, as expected from our above discussion based on the Cauchy-Riemann equations. We illustrate the two (blue and red dot in Fig. 3a) frozen phonon Bloch modes in Fig. 3b, c. They exhibit the expected oscillations versus $z$ with period $2\pi/\text{Re}(k_z) \approx Na = 3a$, with an envelope decaying on a scale much larger than the unit-cell size. Due to its smaller positive imaginary part of $k_z$, the longitudinal mode (blue) has a larger decay length. Due to reciprocity and time inversion symmetry (cf. Fig. 2 and "Methods"), two further eigensolutions with $\text{Re}(k_z) < 0$ arise from flipping the sign, i.e., from $\omega_i(\text{Re}(k_z), \text{Im}(k_z)) = \omega_i(\pm\text{Re}(k_z), \pm\text{Im}(k_z))$.

In our below experiments and calculations on finite-length beams, the solution is a linear superposition of these frozen phonon modes (and a non-Bloch solution for a finite specimen, see below). The prefactors depend on the length of the beam and the boundary conditions. Inside the sample, the different modes interfere, leading to Fabry-Perot-resonator-like interference effects (see below) – in the *static* regime.

Results similar to the ones shown in Fig. 3 for $N = 3$, but for $N = 2$ and $N = 4$, are depicted in Supplementary Fig. 1. The number of local minima increases with increasing $N$. Likewise, more frozen evanescent phonons arise. More broadly, many different metamaterial approaches can lead to local minima in the dispersion relation, likewise leading to anomalous frozen evanescent phonon Bloch modes. This includes mechanism-based metamaterials[41] and others[42]. However, the corresponding effects become only pronounced for very low-frequency local minima, leading to large characteristic exponential decay lengths.

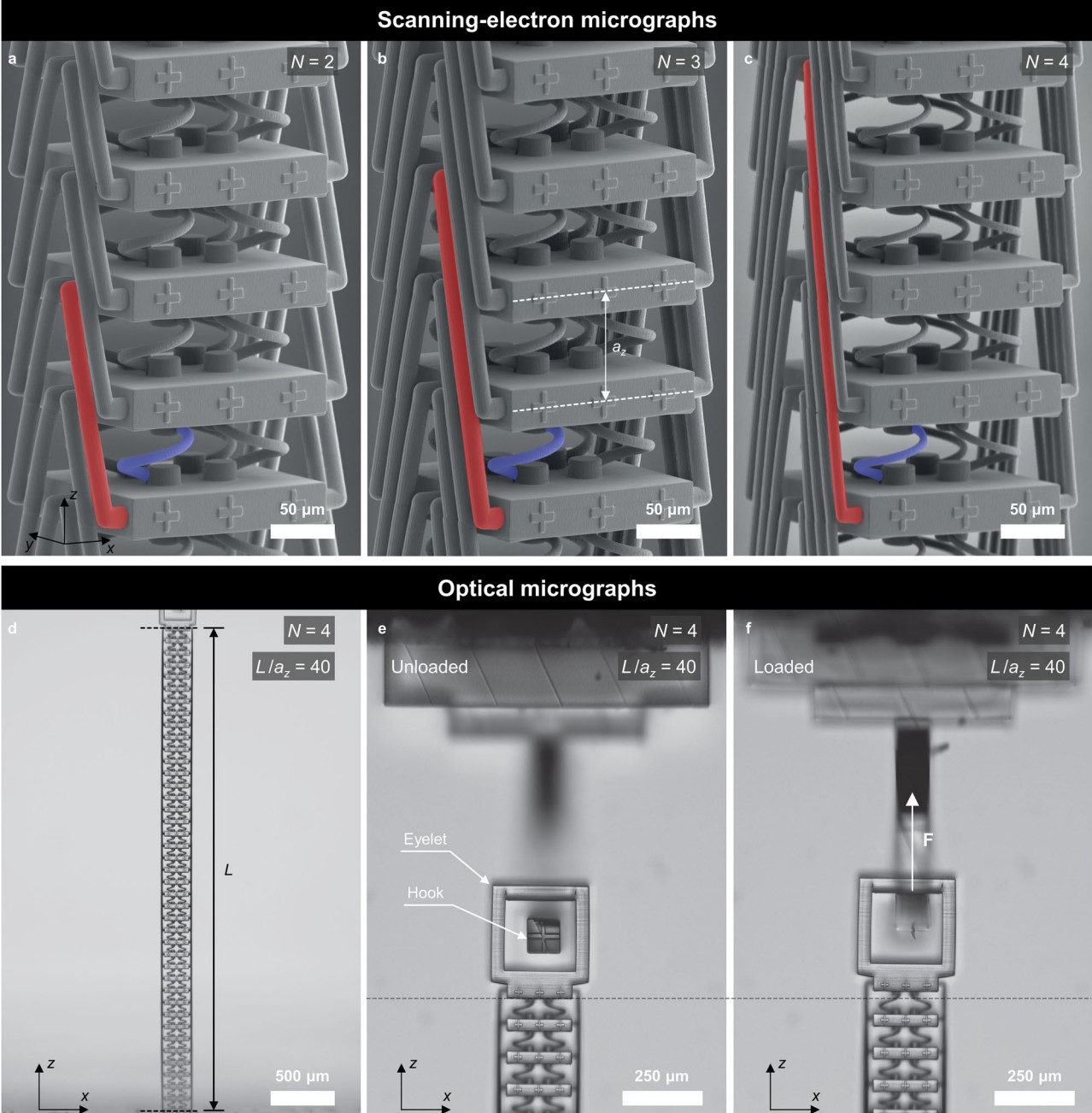

**Fig. 4 | Image gallery of manufactured samples.** Following the blueprint shown in Fig. 2 and for the experiments shown in Fig. 5, we have manufactured a total of 10 different polymer samples on glass substrates with different nonlocal orders of interaction $N = 2, 3, 4$, different relative lengths $L/a_z = 37, 38, 39, 40$, and for realizing two different loading conditions. **a–c** oblique-view scanning-electron micrographs. A spring (rod) mediating the local (nonlocal) interactions is highlighted in blue (red). **d–f** optical micrographs. Panel (**d**) shows the overall sample with length $L$, and panels (**e**) and (**f**) show the hook used for stretching the samples.

## Experiment and theory for finite specimen

We have manufactured polymer samples following the blueprint shown in Fig. 2 by standard 3D laser microprinting (see "Methods"). A gallery of examples for the cases $N = 2, 3, 4$ is shown in Fig. 4. The geometrical parameters of the derived continuum-mechanics model, as well as of the simplified mass-and-spring model, are summarized in the Methods section. The samples additionally contain cross-shaped markers in each unit cell that serve for tracking the mechanical displacement vectors by optical microscopy in a home-built setup, followed by digital image cross-correlation analysis[43,44]. Example optical side-view images for $N = 3$ are shown in Fig. 4d–f and in Fig. 5a. For illustration, Fig. 5b depicts the corresponding simplified mass-and-

spring model. Furthermore, each sample is fixed to the glass substrate at one end, while the other end contains an eyelet, to which a mechanical manipulator can be applied for quasi-statically pulling on the metamaterial beam with total length $L$ along its axis (see Fig. 4, Methods and Supplementary Movies 1–10). We refrain from pushing onto the beams to avoid buckling effects.

Figure 5c summarizes experimental results (orange dots) obtained on a large set of different samples, in which the nonlocal order $N$ and the relative length of the beam samples, $L/a_z$, are systematically varied. In all panels, we plot the $z$-displacement component $u_z$ versus the normalized integer sample coordinate $z/a_z$. For all cases, the total displacement of the beam is in the range of

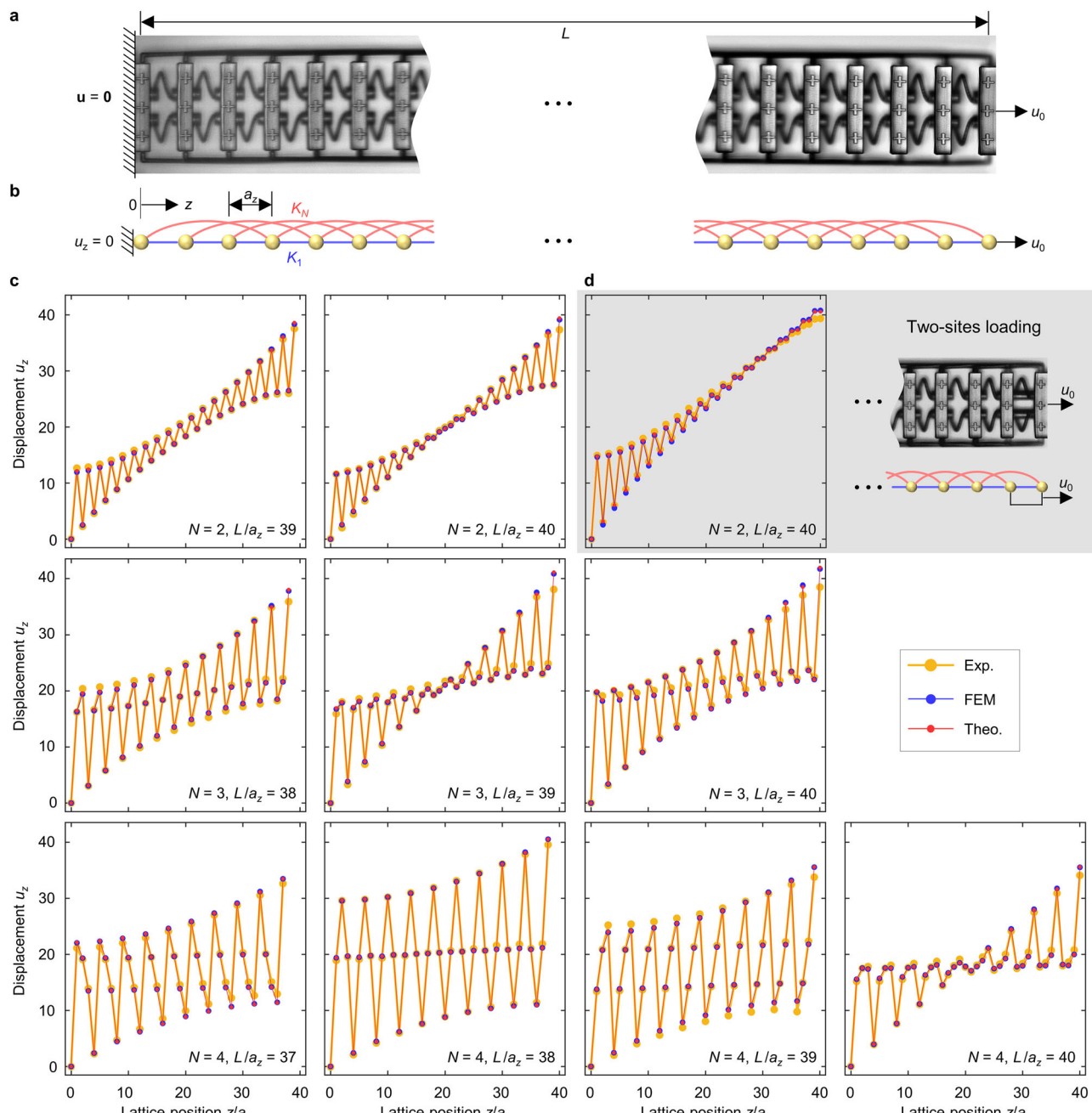

**Fig. 5 | Anomalous displacement fields of stretched metamaterial beams.**
**a** Side-view optical micrograph of a metamaterial beam sample (also see Fig. 2 and Fig. 4) extending from $z = 0$ (glass-substrate side) to $z = L$. **b** Simplified representation in terms of a mass-and-spring model. **c** A force along the $z$-direction is exerted at the last unit cell of the samples at $z = L$, leading to a stretching of the samples along the $z$-direction by an engineering strain of $u_{max}/L \approx 1\%$. The resulting displacement-vector component $u_z$ is recorded as a function of the site number or relative position $z/a_z$. Orange dots refer to experimental measurements, blue dots refer to finite-element numerical calculations, and red dots to solutions of the mass-and-spring model. We note that the finite-element results largely overlap with the other data within the symbol size, indicating excellent agreement. The integer parameters $N$ and $L_z/a$ are indicated in the sub-panels. For an ordinary elastic material, the displacement field would simply follow $u_z(z) = u_{max} z/L$. We rather find pronounced oscillations of $u_z(z)$ that depend on the parameters $N$ and on $L/a_z$. **d** As $N = 2$ and $L/a_z$ in panel (**c**), but the force is applied at the two last unit cells simultaneously.

$u_z(z = L)/L = u_{max}/L \approx 1\%$. Finite-element calculations (blue dots) and solutions of the simple mass-and-spring model (red dots) are shown for comparison. The general qualitative agreement is excellent.

Let us start our discussion of Fig. 5c by recalling that any ordinary (Cauchy) elastic material[15] would simply show a linear displacement field $u_z(z)$ following $u_z(z) = u_{max} z/L$. In sharp contrast, all panels of Fig. 5c rather show additional pronounced large-amplitude spatial oscillations of the displacement field $u_z(z)$ with period $p = N a_z$. This behavior is expected on the basis of our above discussion on frozen evanescent phonons. However, the envelope of the oscillations depends on the relative length of the beam $L/a_z$ as well as on the loading condition. This behavior results from the fact that the static solution is a linear superposition of all frozen-phonon Bloch eigen-modes and of the general non-Bloch solution for a finite-size sample $u_z^{nB}(z) = c_1 + c_2 z$, with constants $c_1$ and $c_2$. The latter is always a solution because the underlying equation of motion contains only up to the second-order spatial derivative. The oscillations of $u_z(z)$ versus $z$ lead to regions of negative slope. This means that the structure is locally

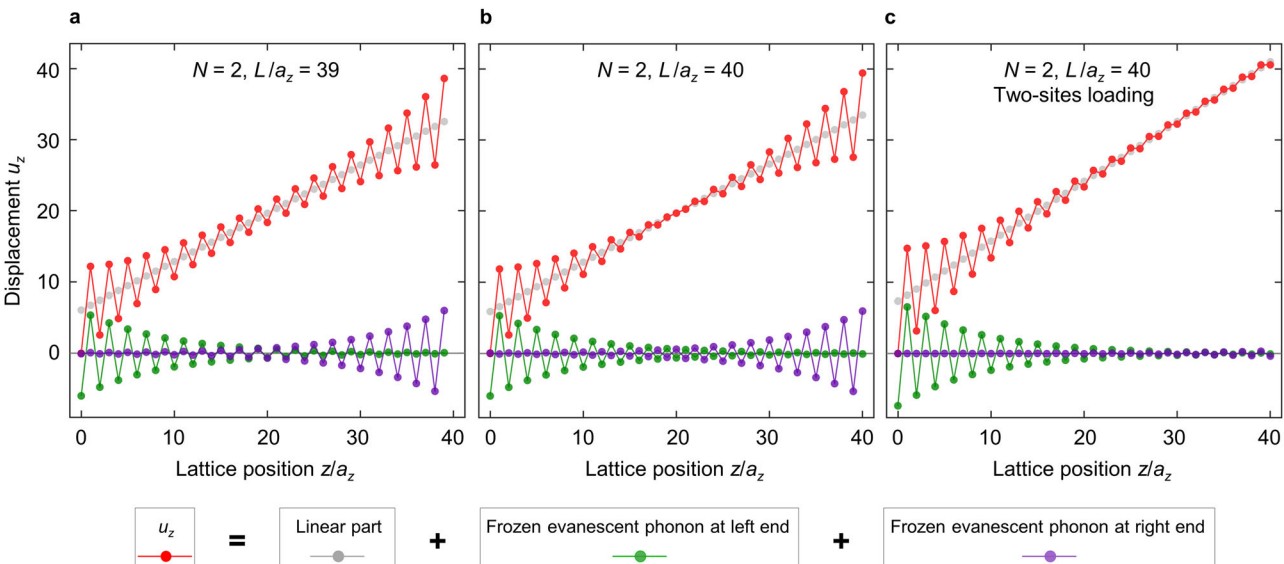

**Fig. 6 | Decomposition of displacement field. a** Result for the case of $N = 2$ and $L/a_z = 39$ in Fig. 5c. The total displacement (cf. red dots) is the solution of the mass-and-spring model shown in Fig. 5b. It is composed of a non-Bloch part $u_z^{nB} = c_1 + c_2 z$ (cf. gray dots), a frozen evanescent phonon eigensolution localized to the left end (cf. green dots), and a frozen evanescent phonon eigensolution localized to the right end (cf. purple dots) of the finite-length beam, respectively. See Methods for details on the decomposition. The two frozen evanescent phonon eigensolutions exhibit in-phase displacement fields in the middle of the beam, leading to constructive interference in the middle (cf. red dots). The straight lines connecting the dots are merely a guide to the eye. **b** Same as (**a**) but for $N = 2$, $L/a_z = 40$. Here, the total displacement (cf. red dots) in the middle shows a smaller oscillation amplitude than in (**a**) due to the destructive interference of the two frozen evanescent phonon eigensolutions. **c** Same as (**a**), but for $N = 2$, $L/a_z = 40$ under the two-site loading condition (cf. Fig. 5d). This boundary condition significantly suppresses the frozen evanescent phonon to the right end of the beam. See Supplementary Figs. 2 and 3 for other examples ($N = 3, 4$) corresponding to Fig. 5c.

compressed in these regions – despite the fact that the structure is elongated overall.

We start by discussing the dependence on $L/a_z$. If the integer $L/a_z$ is commensurable with the integer $N$, which holds true for the three cases $L/a_z = 40$ and $N = 2$, $L/a_z = 39$ and $N = 3$, and $L/a_z = 40$ and $N = 4$ in Fig. 5c, the envelope of the oscillation exhibits a constriction in the middle of the beam. In these cases, the frozen evanescent phonon modes decaying from the left and from the right-hand side of the beam have the same amplitude, yet opposite sign, leading to destructive interference in the middle of the beam. For non-commensurable combinations of the two integers $L/a_z$ and $N$, the envelope of the oscillations is distinctly different. Hence, one does not get destructive interference. This behavior somewhat resembles Fabry-Perot resonances, for which resonance occurs whenever the cavity length is an integer multiple of half the wavelength. We emphasize, though, that we here discuss the static regime. To support this interpretation, we have decomposed the displacement field of the mass-and-spring model into different parts, including a linear (affine) solution and frozen evanescent phonon solutions (see "Methods"). As an example, we show in Fig. 6 the results for $N = 2$ corresponding to Fig. 5c, d. As explained above, the two evanescent phonons (cf. green and purple dots) decay toward the center of the system. They both exhibit an oscillation period of $Na_z = 2a_z$. In addition, the two evanescent phonons have the same amplitude, and they are out-of-phase due to the mirror-reflection symmetry of the system. Therefore, for the commensurable case (cf. Fig. 6b), the two evanescent phonons destructively interfere in the middle of the beam. For the incommensurable case (cf. Fig. 6a), constructive interference occurs in the middle. Corresponding results for $N = 3$ and $4$ are shown in Supplementary Figs. 2 and 3.

The behavior would clearly be equivalent if, for a fixed sample with $L/a_z = 40$ and $N = 4$, we had applied the force at the unit cell at $z/a_z = 39$ instead of $z/a_z = 40$. This means that the observed behavior is a violation of Saint Venant's principle[14], which essentially states that moving the point at which a force is applied by a small amount should lead to only small changes of the displacement field far away from this point[15]. The reason for this violation lies in the anomalously large decay length of frozen evanescent phonons that carries a change at one point to remote points. This behavior could be used for remote sensing of where in space a mechanical stimulus is applied.

We further study the influence on the loading conditions. For all cases discussed thus far in Fig. 5c, the loading force (apart from the fixed substrate side) has been applied only to one unit cell of the beam. In Fig. 5d (highlighted in gray) results for applying the loading force to the last as well as to the second-last unit cell are shown. We accomplish this situation by mechanically connecting these two cells during the 3D manufacturing process (see inset). The displacement $u_z(z)$ shown in Fig. 5d is distinctly different from the behavior right next to it on the left in that no oscillations occur at that end of the beam at which the force is applied to two unit cells at once. Here, the boundary conditions simply lead to a negligibly small amplitude of the frozen evanescent phonon mode decaying away from that end, as can be seen from the decomposed displacement field shown in Fig. 6c.

The described dependencies on the integers $L/a_z$ and $N$ result from different forces acting within the sample. One would therefore expect, for example, that the effective Hooke's spring constant of the beam non-monotonously depends on $L/a_z$ for fixed $N = 2, 3, 4$ rather than only scaling like $\propto 1/L$ for an ordinary elastic beam. Supplementary Fig. 4 shows that this is indeed the case.

Following our introduction (cf. Fig. 1), nonlocality is not the only means to arrive at large-decay-length frozen evanescent phonons. In Supplementary Fig. 5, we discuss a mechanism-based metamaterial[41] as another example. A local minimum of the dispersion band occurs at the edge of the first Brillouin zone. This local minimum and the resulting frozen evanescent phonons are connected to a floppy mode of the mechanism-based metamaterial with ideal-hinge connections. The authors[41] described a route toward obtaining large characteristic lengths in mechanics that did not involve band structures or (evanescent) Bloch modes. Our approach shades a different light onto these periodic metamaterials rather based on their Bloch eigenstates.

Our discussion so far has been restricted to effectively one-dimensional decay problems for which one local minimum in the dispersion relation $\omega_i(k)$ leads to four frozen evanescent phonon modes per branch $i$ (see Figs. 1b, 3a). In effectively two-dimensional problems (see Supplementary Fig. 6), we find infinitely many frozen evanescent eigensolutions at $\omega = 0$, forming an area within the plane spanned by $\mathrm{Re}(k_x)$ and $\mathrm{Re}(k_y)$ (see Supplementary Fig. 7). For example, when exerting a point-like force onto a two-dimensional thin square-shaped membrane fixed at all of its four edges, the membrane's static out-of-plane displacement spatially oscillates within the plane (cf. Supplementary Fig. 8). These pronounced spatial oscillations also occur in the lowest-frequency eigenmodes of the finite membrane (see Supplementary Figs. 6 and 7), including the fundamental mode, which usually exhibits only a single maximum in the middle of the membrane. Here, we move upwards on the evanescent branch starting from the black dot at $\omega = 0$ in Fig. 1b. The modes can be excited by time-harmonic excitation at frequencies close to the membrane eigenfrequencies (cf. Supplementary Fig. 8).

## Discussion

Frozen evanescent waves are the Bloch eigensolutions of any linear and infinitely periodic problem for complex-valued wavenumbers at zero frequency. Using the Cauchy-Riemann equations, we have established a connection between the imaginary part of their wavenumber, which is the inverse of their characteristic exponential decay length, and the frequency at a local minimum of the ordinary (real-valued) wave's dispersion relation. This approach allows us to connect to the static behavior of finite-size samples and to systematically determine all characteristic length scales of a problem. This has not been possible previously for elastic waves (phonons), which we have used as an example. Our results are also directly applicable to other types of waves, including electromagnetic and acoustic waves.

## Methods
### Calculation of complex phonon band structures
We compute the phonon band structure for complex-valued wavenumbers by solving the eigenvalue equation

$$-\rho\omega^2 u_\alpha(x_\tau) = -k_\eta k_\xi C_{\alpha\xi\beta\eta} u_\beta(x_\tau) + \frac{\partial}{\partial x_\xi}\left(C_{\alpha\xi\beta\eta}\frac{\partial u_\beta(x_\tau)}{\partial x_\eta} + \mathrm{i}k_\eta C_{\alpha\xi\beta\eta} u_\beta(x_\tau)\right)$$
$$+ \mathrm{i}k_\xi C_{\alpha\xi\beta\eta}\frac{\partial u_\beta(x_\tau)}{\partial x_\eta} \tag{5}$$

together with the stress-free boundary condition

$$n_\xi\left(C_{\alpha\xi\beta\eta}\frac{\partial u_\beta(x_\tau)}{\partial x_\eta} + \mathrm{i}k_\eta C_{\alpha\xi\beta\eta} u_\beta(x_\tau)\right) = 0, \tag{6}$$

and the periodic boundary condition

$$u_\beta(x_\tau) = u_\beta(x_\tau + a_\tau). \tag{7}$$

Herein, $\rho$ and $C_{\alpha\xi\beta\eta}$ refer to the mass density and the fourth-rank Cauchy elasticity tensor. For isotropic elastic materials, the elasticity tensor can be expressed as $C_{\alpha\xi\beta\eta} = (E\nu/((1+\nu)(1-2\nu))\delta_{\alpha\xi}\delta_{\beta\eta} + E/(2+2\nu)\left(\delta_{\alpha\beta}\delta_{\xi\eta} + \delta_{\alpha\xi}\delta_{\beta\xi}\right)$, with $E$ and $\nu$ being the Young's modulus and the Poisson's ratio, respectively. The above Eqs. (5) – (7) results from the Bloch ansatz for the displacement, $u_\beta(x_\tau)\exp\left(\mathrm{i}(\omega t - k_\xi x_\xi)\right)$ with $u_l(x_\tau)$ being a periodic function. $u_l(x_\tau)$, $\omega$ and $k_\xi$ are the eigen-displacement, the eigenfrequency, and the Bloch wavenumber, respectively. The spatial coordinate is denoted by $x_\tau$, and the lattice vector is represented by $a_\tau$. All the indices, $\alpha, \beta, \xi, \eta$, and $\tau$ run from 1 to 3 for the three-dimensional elasticity problem considered. For two-dimensional problems, they run from 1 to 2. Repeated indices follow the Einstein summation convention.

We implement Eqs. (5) – (7) within the commercial software Comsol Multiphysics by using its partial differential equation (PDE) module. To obtain the complex phonon band structures (cf. Fig. 3 and Supplementary Figs. 1 and 5) in the main paper, we apply periodic boundary conditions along the $z$-direction and treat all other boundaries as stress-free. Therefore, the Bloch wavevector is given by $\mathbf{k} = (0, 0, k_z)$. The unknown wavenumber, $k_z$, is solved from the above Eqs. (5) – (7) with pre-described real-valued frequencies. In all calculations, we assume the following elastic parameters for the constitutive material: Young's modulus $E = 4.19\,\mathrm{GPa}$, Poisson's ratio $\nu = 0.3$, and mass density $\rho = 1190\,\mathrm{kg/m^3}$.

For a real-valued mass density $\rho$ and a real-valued elasticity tensor $C_{\alpha\xi\beta\eta}$, the calculated complex phonon bands show two mirror-symmetry planes, i.e., $\mathrm{Re}(\omega)$-$\mathrm{Re}(k_z)$ and $\mathrm{Re}(\omega)$-$\mathrm{Im}(k_z)$. As a result, for clarity, we only plot that part of the bands with $\mathrm{Re}(k_z) > 0$ (cf. Fig. 3 and Supplementary Figs. 1 and 5).

The above two mirror symmetries of the complex-valued phonon bands result from the following two symmetry properties: (1) $\overline{\omega^2(k_\xi)} = \omega^2\left(-\overline{k_\xi}\right)$, (2) $\omega^2(-k_\xi) = \omega^2(k_\xi)$. Herein, $\overline{\omega^2(k_\xi)}$ and $\overline{k_\xi}$ denote the complex conjugate of $\omega^2(k_\xi)$ and $k_\xi$, respectively. The symmetry property (1) simply follows from taking the complex conjugate of Eqs. (4) – (7) if the material parameters, as $\rho$ and $C_{\alpha\xi\beta\eta}$, are real-valued. This condition also implies time-inversion symmetry[15]. We remark that symmetry (2) becomes identical to symmetry (1) for real wavenumbers. Next, we briefly show that symmetry (2) applies to complex-valued wavenumbers as well.

Suppose that $u_\alpha(x_\tau)$, $\omega$, and $k_\xi$ correspond to a solution of the above eigenvalue Eqs. (4) – (7) and that $v_\alpha(x_m)$, $\omega'$, and $-k_\xi$ constitute another solution. Therefore, $v_\alpha(x_\tau)$, $\omega'$, and $-k_\xi$ must satisfy the below equation

$$-\rho\omega'^2 v_\alpha(x_\tau) = -k_\eta k_\xi C_{\alpha\xi\beta\eta} v_\beta(x_\tau) + \frac{\partial}{\partial x_\xi}\left(C_{\alpha\xi\beta\eta}\frac{\partial v_\beta(x_\tau)}{\partial x_\eta} - \mathrm{i}k_\eta C_{\alpha\xi\beta\eta} v_\beta(x_\tau)\right)$$
$$- \mathrm{i}k_\xi C_{\alpha\xi\beta\eta}\frac{\partial v_\beta(x_\tau)}{\partial x_\eta} \tag{8}$$

together with the stress-free boundary condition

$$n_\xi\left(C_{\alpha\xi\beta\eta}\frac{\partial v_\beta(x_\tau)}{\partial x_\eta} - \mathrm{i}k_\eta C_{\alpha\xi\beta\eta} v_\beta(x_\tau)\right) = 0, \tag{9}$$

and the periodic boundary condition

$$v_\beta(x_\tau) = v_\beta(x_\tau + a_\tau). \tag{10}$$

We multiply both sides of Eq. (8) with $u_\alpha(x_\tau)$ and integrate over the unit cell, and obtain

$$-\omega'^2 \int \rho(v_\alpha(x_\tau)u_\alpha(x_\tau))\mathrm{d}V = \int C_{\alpha\xi\beta\eta}\left(-k_\eta k_\xi v_\beta(x_\tau)u_\alpha(x_\tau) - \frac{\partial v_\beta(x_\tau)}{\partial x_\eta}\frac{\partial u_\alpha(x_\tau)}{\partial x_\xi}\right)\mathrm{d}V$$
$$+ \mathrm{i}\int C_{\alpha\xi\beta\eta}\left(k_\eta v_\beta(x_\tau)\frac{\partial u_\alpha(x_\tau)}{\partial x_\xi} - k_\eta\frac{\partial v_\beta(x_\tau)}{\partial x_i}u_\alpha(x_\tau)\right)\mathrm{d}V. \tag{11}$$

The boundary conditions Eqs. (6), (7), (9), and (10) are used to arrive at the above equation. Likewise, multiplying both sides of Eq. (1) with $v_\alpha(x_\tau)$ leads to

$$-\omega^2 \int \rho(u_\alpha(x_\tau)v_\alpha(x_\tau))\mathrm{d}V = \int C_{\alpha\xi\beta\eta}\left(-k_\eta k_\xi u_\beta(x_\tau)v_\alpha(x_\tau) - \frac{\partial u_\beta(x_\tau)}{\partial x_\eta}\frac{\partial v_\alpha(x_\tau)}{\partial x_\xi}\right)\mathrm{d}V$$
$$- \mathrm{i}\int C_{\alpha\xi\beta\eta}\left(k_\eta u_\beta(x_\tau)\frac{\partial v_\alpha(x_\tau)}{\partial x_\xi} - k_\xi\frac{\partial u_\beta(x_\tau)}{\partial x_\eta}v_\alpha(x_\tau)\right)\mathrm{d}V. \tag{12}$$

It can be shown that the right-hand sides of Eqs. (11) and (12) are equal due to the major symmetry of the elasticity tensor, i.e., via $C_{\alpha\xi\beta\eta} = C_{\beta\eta\alpha\xi}$. By subtracting Eq. (12) from Eq. (11), we have

$$\left(\omega^2 - \omega'^2\right) \int \left(\rho u_\alpha(x_\tau) v_\alpha(x_\tau)\right) dV = 0. \quad (13)$$

In general, the integrand in Eq. (13) is not zero. The two eigenfrequencies are identical, $\omega'^2 = \omega^2$, or alternatively, $\omega^2(-k_\xi) = \omega^2(k_\xi)$. This relation holds true even for lossy materials with complex-valued elasticity tensors. In fact, the symmetry of $C_{\alpha\xi\beta\eta} = C_{\beta\eta\alpha\xi}$ can be deduced from the requirement of reciprocity[15].

### Static simulations of metamaterial beams

We simulate the stretching of the metamaterial beam by using the commercial software Comsol Multiphysics with the following static elasticity equation

$$\frac{E}{2(1+v)(1-2v)} \frac{\partial}{\partial x_\alpha}\left(\frac{\partial u_\beta(x_m)}{\partial x_\beta}\right) + \frac{E}{2(1+v)} \frac{\partial}{\partial x_\beta}\left(\frac{\partial u_\alpha(x_m)}{\partial x_\beta}\right) = 0, \quad (14)$$

where, $\rho$, $E$ and $v$ are the mass density, the Young's modulus, and the Poisson's ratio of the constitutive material, respectively. In the static calculation, we use the same elasticity parameters as in the calculation of the complex-valued phonon band structures. We apply fixed boundary conditions to one end of the metamaterial beam and prescribe a finite displacement at the other end.

The spring constants, $K_1$ and $K_N$, used in the simplified mass-and-spring model for the metamaterial beam, are also obtained by solving Eq. (14). Here, two plates with only nearest-neighbor connections or only third-nearest-neighbor connections, respectively, are modeled to derive the parameters $K_1$ and $K_N$. One plate is fixed at the bottom, while the other plate is loaded with force $F_z$ along the axial direction. We obtain the axial displacement $u_z$ of the loaded plate from Comsol Multiphysics and derive the effective Hooke's spring constant $F_z/u_z$. The obtained stiffness parameters for $N = 2$, 3, and 4 are $K_1 = 33.5$ N/m, $K_N = 2562.9$ N/m, $K_1 = 33.5$ N/m, $K_N = 2401.3$ N/m, and $K_1 = 8.1$ N/m, $K_N = 756.7$ N/m, respectively. The effective Hooke's spring constants of metamaterial beams with finite length (cf. Supplementary Fig. 4) are derived analogously.

### Decomposition of displacement field

For the simplified finite-length mass-and-spring model in Fig. 5b, we denote the axial ($z$-direction) displacement of each mass by $u_n$ with $n = 0, 1, 2 \ldots, L/a_z$. We first focus on the masses in the bulk, i.e., $n = N, N+1, \ldots L/a_z - N$. Each of these masses is connected to two immediate neighbors and two $N$-th nearest neighbors on both sides. Thus, the force-balance equation for the mass $n$ reads:

$$K_1(u_{n+1} + u_{n-1} - 2u_n) + K_N(u_{n+N} + u_{n-N} - 2u_n) = 0, n = N, N+1, \ldots L/a_z - N. \quad (15)$$

Apparently, this set of equations always supports the linear non-Bloch solution

$$u_n^{nB} = C_1 + C_2 n, n = 0, 1, 2 \ldots, L/a_z, \quad (16)$$

$C_1$ and $C_2$ are constants. Additionally, multiple frozen evanescent phonons of the following form are possible

$$u_n^j = \exp\left(ik_j(na - L/2)\right), n = 0, 1, 2 \ldots, L/a_z. \quad (17)$$

Here, the complex wavenumber $k_j$ satisfies the following condition

$$K_1\sin^2\left(\frac{k_j a}{2}\right) + K_N\sin^2\left(N\frac{k_j a}{2}\right) = 0. \quad (18)$$

Equation (18) has infinitely many solutions for $k_j$. Here, we only need to consider $k_j$ with its real part inside of the first Brillouin zone, $-\pi/a_z < \text{Re}(k_j) \leq \pi/a_z$, as other solutions are simply shifted by an integer multiple of $2\pi/a_z$, which does not influence the displacement solution Eq. (17).

For the considered parameter settings in Fig. 5c, we have the following solutions for $k_j$:

$$N = 2, k_1 \approx (1 + 0.0364i)\pi/a_z, k_2 \approx (1 - 0.0364i)\pi/a_z \quad (19)$$

$$N = 3, k_1 \approx (0.6662 + 0.0217i)\pi/a_z, k_2 \approx (-0.6662 + 0.0217i)\pi/a_z,$$
$$k_3 = -k_1, k_4 = -k_2. \quad (20)$$

$$N = 4, k_1 \approx (0.4998 + 0.0116i)\pi/a_z, k_2 \approx (-0.4998 + 0.0116i)\pi/a_z,$$
$$k_3 = -k_1, k_4 = -k_2,$$
$$k_5 \approx (1 + 0.0164i)\pi/a_z, k_6 \approx (1 - 0.0164i)\pi/a_z \quad (21)$$

The displacement field in Eq. (17) for $k_j$ with a positive imaginary part represents a frozen evanescent phonon that exponentially decays to the right, while the one with a negative imaginary part stands for a frozen evanescent phonon exponentially decaying to the left.

The general displacement field of the finite-length system in Fig. 5b is a linear combination of the above solutions, i.e.,

$$u_n = C_1 + C_2 n + \sum_j E_j \exp\left(ik_j na\right) \quad (22)$$

The displacement field Eq. (22) automatically ensures the force-balance equation Eq. (15) for the masses in bulk, i.e., $n = N, N+1, \ldots L/a_z - N$. The unknown complex coefficients, $C_1, C_2$, and $E_j$, can be determined from the boundary conditions applied to the mass-and-spring chain.

For the single-site loading as in Fig. 5c, the left most mass and the right most mass have prescribed displacements, $u_0 = 0$, and $u_{L/a_z} = u_{\max}$. In addition, masses $n = 1, \ldots N-1$, and masses $n = L/a_z - N+1, \ldots L/a_z - 1$ must be in force-balance, too. For the two-sites loading condition in Fig. 5d, the mass $n = L/a_z - 1$ is similarly prescribed with displacement, $u_{L/a_z-1} = u_{\max}$. Other conditions are the same as the single-site loading conditions. It can be easily verified that the number of the boundary conditions $2N$, exactly matches the number of the unknown coefficients. Thus, the unknown coefficients can be determined. Afterward, the total displacement field Eq. (22), as well as individual parts, including the linear part and the frozen evanescent phonons, can be obtained. The analytical formula for the unknown coefficients is very lengthy and thus omitted here.

### Metamaterial-beam fabrication

We model the metamaterial beam in the commercial software Comsol Multiphysics and export the STL file for fabrication. Due to multiple overhanging beams in the metamaterial model, the geometry is split into different parts and printed sequentially. Geometry compensations are considered to minimize the discrepancy between the fabricated samples and the targeted geometry parameters. We fabricate the metamaterial samples by using a commercial 3D laser printer (Professional GT, Nanoscribe GmbH) with a 25 × microscope objective lens

(numerical aperture NA = 0.8, Carl Zeiss) at a laser focus scanning speed of 0.115 m/s and a printing laser power of 25 mW. A commercial liquid photoresist (IP-S, Nanoscribe GmbH) was used, and the dip-in printing mode was selected. We printed the metamaterial beam vertically on a glass substrate coated with indium tin oxide and silanized using 3-(trimethoxysilyl)propylmethacrylate. The hatching distance and slicing distance were chosen to be $0.3 \mu m$ and $0.5 \mu m$, respectively. A bottom plate with a loading hook was printed separately with the printing parameters of 50 mW for the laser power, 0.140 m/s for the focus scanning speed, $0.5 \mu m$ and $1.5 \mu m$ for the hatching distance and slicing distance of the baseplate. For the hook, $0.3 \mu m$ and $0.5 \mu m$ were chosen. On top of the metamaterial beam, an auxiliary eyelet was printed to ease the connection between the metamaterial and the loading hook. More details about the printing setting can be found in the GWL files that are published in [https://doi.org/10.35097/RgiKSjVUuivXDyKi].

After completing the printing job, we first immersed the sample in propylene glycol methyl ether acetate and ethanol for 20 min each to remove excess photoresist. Afterward, the samples were supercritically dried in $CO_2$ (Leica EM CPD 300).

## Metamaterial beam stretching experiment

In the experiment, one end of the metamaterial beam is fixed on a bottom glass substrate while the other end is connected via the printed eyelet and the loading hook to a computer-controlled motorized actuator (TRA25CC, Newport). Here the hook was glued (Plast Special, UHU) onto a stamp screwed to the actuator and made from aluminum. Beginning with the upstretched initial state of the metamaterial beam, the actuator is programmed to gradually move to a prescribed displacement $u_{max} \approx 0.01 L$ and then stay steady for 2 seconds. Afterward, the actuator is again programmed to gradually move backward to its initial position (see Supplementary Movies 1–10). We image the metamaterial beam from the side with a digital camera (Blackfly BFLY-PGE-50H5C, Point Gray Research) and record the complete loading-unloading process at a frame rate of 7.5 fps. We extract the displacement of each unit cell from the cross-shaped markers by using a digital-image cross-correlation algorithm[43]. For each metamaterial sample, we repeat the loading-unloading cycle ten times with a waiting time of 30 s in between the cycles. The obtained displacements are averaged over ten measurements to derive the data points shown in Fig. 5. More details about the measurement and raw data can be found in the data repository published together with this work [https://doi.org/10.35097/RgiKSjVUuivXDyKi].

## Experimental determination of effective Hooke's spring constants

In the above stretching experiment, we also record the force applied to the metamaterial beam end by using a force sensor (K3D40, ME-Meßsysteme GmbH). With the displacement data for the top unit cell of the metamaterial beam (derived from the above digital-image cross-correlation algorithm), we obtain the force-displacement curve. The effective Hooke's spring constant is derived by fitting the slope of the force-displacement curve using a least square algorithm. As mentioned above, we repeat the loading-unloading cycle ten times for each metamaterial beam sample. Therefore, ten effective Hooke's spring constants, with each corresponding to one loading-unloading cycle, are derived. We obtain the averaged effective Hooke's spring constant and the standard deviation from the ten results. The raw displacement data and force data can be found in the data repository published together with this work [https://doi.org/10.35097/RgiKSjVUuivXDyKi].

## Data availability

The simulation and experiment data generated in this study are available from the corresponding authors upon request and are

published in the open-access data repository of the Karlsruhe Institute of Technology [https://doi.org/10.35097/RgiKSjVUuivXDyKi].

## Code availability

The code that produces the results and the plots within the paper are available from the corresponding authors upon request and are published in the open-access data repository of the Karlsruhe Institute of Technology [https://doi.org/10.35097/RgiKSjVUuivXDyKi].

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

## Acknowledgements
We acknowledge financial support by the Deutsche Forschungsgemeinschaft (DFG, German Research Foundation) under Germany's Excellence Strategy via the Excellence Cluster "3D Matter Made to Order", EXC-2082/1-390761711, by the Carl Zeiss Foundation through the "Carl-Zeiss-Foundation-Focus@HEiKA", by the State of Baden-Württemberg, by the Helmholtz program "Materials Systems Engineering", by the Alexander von Humboldt Foundation. M.K. is grateful for support from the ANR PNanoBot (ANR-21-CE33-0015) and the ANR OPTOBOTS project (ANR-21-CE33-0003). K.W. acknowledges support from the China Scholarship Council (CSC). P.S. Thanks to the Hector Fellow Academy for their support.

## Author contributions
Theory analysis and numerical calculations were performed by Y.C., K.W., and M.K. Experiments were conducted by J.LG.S. Electron micrographs were taken by J.LG.S., P.S., and S.K. M.W. and Y.C. drafted the paper. All authors contributed to the interpretation of data, discussion, and revision of the paper. M.W. supervised the overall project.

## Funding

## Competing interests
The authors declare no competing interests.
