## [Peer Review File · Nature Communications]

REVIEWER COMMENTS

Reviewer #1 (Remarks to the Author):

The authors study zero frequency spatially decaying modes in an effectively 1D mechanical system with non monotonic band dispersion (due to nonlocal springs), and show that the decay length can be engineered to large. They further show that these spatially decaying modes have significant effect in the deformation of the system under static loading. While the concept of evanescent modes in the bulk gap and the governing decay length equation in the article is not new (well-known Jackiw-Rebbi (JR) analysis also predicts the same decay length as well as spatial profile. See, for example, Alisepahi et al. Commun. Phys. 6, 334 (2023). Even though, in that article, the JR analysis was used for high frequency bulk gap opening, same analysis can be used for gap opening at zero frequency of this article), the effect of these modes in static deformation of the structure and the resulting break down of Saint Venant's principle are new and interesting.

Having said that, I feel the theoretical analysis in explaining the observations of Fig. 5 is slightly lacking. The authors match the experimental results with solutions obtained from an effective spring mass model. However, the direct connection between the evanescent modes and the deformation of the system could be made clearer. More specifically, if the deformation pattern is a linear combination of an affine deformation (z/L) and the four evanescent modes (let me call them u_i , $i = 1, 2, 3, 4$), then deformation can be written as $u = a*z/L + \sum_{i=1}^4 b_i u_i$, where a and b_i are unknown coefficients. Then satisfying the boundary conditions, one can directly obtain these coefficients. This calculation will shed direct light on the difference between systems commensurate and incommensurate lengths (at the moment the difference between these two cases seems to be just an observation without an explicit proof).

The rest of the arguments of the paper are well justified in my opinion. With the addition suggested above, I suggest publication of this article in Nature Communications.

Reviewer #2 (Remarks to the Author):

This manuscript introduces frozen evanescent phonons as eigenstates of the phonon band structure characterized by complex-valued wavenumbers that freeze at zero frequency. It commences with a thorough discussion of the analytical nature of the phonon band structure, elucidating how extrema in the band structure correspond to the presence of evanescent modes. The study further investigates the influence of boundary conditions and specimen size on the response of finite-size materials to external forces, uncovering notable violations of Saint Venant's principle. Additionally, the manuscript provides mechanism-based metamaterials as an alternative approach for generating local minima in the supplemental materials and also offers insights into the two-dimensional case. The findings of this manuscript are intriguing, and the comprehensive evidence provided through experimental simulations and theoretical analyses adds substantial weight to the conclusions. I would recommend potential publication once the addressed concerns have been appropriately resolved.

1. At the extrema of the band ω_i , the Taylor expansion ignores the 1st derivative term, is it because near the extrema the quadratic characteristic dominates? In line 72, the author calculates ζ from the 2nd derivative term. Should there be a $1/2$ factor included? as the coefficient of the 2nd derivative term should be $f''/2$.
2. I don't see the finite element data in Figure 5. Were they forgotten to be added or are they being covered by the experimental and theoretical data?
3. Is it possible to have multiple local minima in the 1D system? If so, what will the response look like when applying quasi-static loading?
4. In Supplementary Fig. 3, the authors illustrate anomalous frozen evanescent modes in mechanism-based mechanical metamaterials. It appears that the mode is the floppy mode. Does this imply that any mechanical metamaterials with floppy modes have the potential to realize this effect? Could the authors provide insights into the connection between the floppy mode and frozen evanescent phonons? Additionally, it seems that the red dot is missing in panel b.
5. The authors present the eigenmodes for the nonlocal model in Supplementary Fig. 5, which display pronounced oscillations with a spatial period of $3a$. To observe these frozen evanescent modes, what type of excitation should be applied, and where should it be applied? Should it be a quasi-static loading similar to the 1D case, or should it be a load with the real part of the eigenfrequency?

Response to Reviewers

We thank the reviewers for their response. In what follows, *we repeat their comments in red and italics*, we respond in green, we cite from the original manuscript in black, and highlight changes made to the manuscript in blue.

Reviewer #1

The authors study zero frequency spatially decaying modes in an effectively 1D mechanical system with non monotonic band dispersion (due to nonlocal springs), and show that the decay length can be engineered to large. They further show that these spatially decaying modes have significant effect in the deformation of the system under static loading. While the concept of evanescent modes in the bulk gap and the governing decay length equation in the article is not new (well-known Jackiw-Rebbi (JR) analysis also predicts the same decay length as well as spatial profile. See, for example, Alisepahi et al. Commun. Phys. 6, 334 (2023). Even though, in that article, the JR analysis was used for high frequency bulk gap opening, same analysis can be used for gap opening at zero frequency of this article), the effect of these modes in static deformation of the structure and the resulting break down of Saint Venant's principle are new and interesting.

We are grateful to reviewer #1 for their positive assessment of our manuscript, especially for stating that “the effect of these modes in static deformation of the structure and the resulting break down of Saint Venant's principle are new and interesting”.

We agree with reviewer #1 in that the Jackiw-Rebbi analysis has been applied for studying evanescent modes at finite frequencies. We respectfully emphasize that it could possibly have been applied to the static case – but it has not been applied to the static case. Our paper specifically addresses the static case. In more detail, the Jackiw-Rebbi solution has been applied to finite-frequency modes localized at an interface separating two distinct topological phases. We are not discussing two distinct topological phases. Nevertheless, it is interesting for the reader to realize a connection between frozen evanescent phonons and the Jackiw-Rebbi solution. Therefore, we have cited the paper addressed by reviewer #1 in our Introduction:

“...

Ordinarily, evanescent waves are associated with *finite frequencies inside frequency bandgaps* [6]. A well-known example is the Jackiw-Rebbi solution [7-9] localized at a domain-wall separating two different topological phases. Here, we apply the concept of evanescent waves in single-domain samples to the *static* regime, in which the wave gets “frozen” in time. These special Bloch modes allow for making a direct connection between unusual static and dynamic properties, which has previously been unclear, hampering the rational design of unusual static behavior in metamaterials. The Bloch eigenmodes of elastic waves in metamaterials, phonons, serve as an example. By a general discussion based on the Cauchy-Riemann equations, treating the band structure as an analytical function, we introduce the concept of frozen evanescent phonons.... “

The references [7-9] are new and include the paper suggested by reviewer #1. We have re-numbered all references accordingly.

Having said that, I feel the theoretical analysis in explaining the observations of Fig. 5 is slightly lacking. The authors match the experimental results with solutions obtained from an effective spring mass model. However, the direct connection between the evanescent modes and the deformation of the system could be made clearer. More specifically, if the deformation pattern is a linear combination of an affine deformation (z/L) and the four evanescent modes (let me call them u_i , $i = 1, 2, 3, 4$), then deformation can be written as $u = a \cdot z/L + \sum_{i=1}^4 b_i u_i$, where a and b_i are unknown coefficients. Then satisfying the boundary conditions, one can directly obtain these coefficients. This calculation will shed direct light on the difference between systems commensurate and incommensurate lengths (at the moment the difference between these two cases seems to be just an observation without an explicit proof).

We thank reviewer #1 for this very insightful comment. Following this comment, we have decomposed the displacement field into the frozen-phonon eigensolutions. The results are presented in the new Fig. 6 in the main paper. In addition, we have added further results in the two new Supplementary Figs. 2 and 3. We refer to these figures in the discussion on Fig. 5 in the main paper:

“... ”

We start by discussing the dependence on L/a_z . If the integer L/a_z is commensurable with the integer N , which holds true for the three cases $L/a_z = 40$ and $N = 2$, $L/a_z = 39$ and $N = 3$, and $L/a_z = 40$ and $N = 4$ in Fig. 5c, the envelope of the oscillation exhibits a constriction in the middle of the beam. In these cases, the frozen evanescent phonon modes decaying from the left and from the right-hand side of the beam have the same amplitude, yet opposite sign, leading to destructive interference in the middle of the beam. For non-commensurable combinations of the two integers L/a_z and N , the envelope of the oscillations is distinctly different. Hence, one does not get destructive interference. This behavior somewhat resembles Fabry-Perot resonances, for which a resonance occurs whenever the cavity length is an integer multiple of half the wavelength. We emphasize though that we here discuss the static regime. To support this interpretation, we have decomposed the displacement field of the mass-and-spring model into different parts, including a linear (affine) solution and frozen evanescent phonon solutions (see Methods). As an example, we show in Fig. 6 the results for $N = 2$ corresponding to Fig. 5c and 5d. As explained above, the two evanescent phonons (cf. green and purple dots) decay towards the center of the system. They both exhibit an oscillation period of $Na_z = 2a_z$. In addition, the two evanescent phonons have the same amplitude and they are out-of-phase due to mirror-reflection symmetry of the system. Therefore, for the commensurable case (cf. Fig. 6b), the two evanescent phonons destructively interfere in the middle of the beam. For the incommensurable case (cf. Fig. 6a), constructive interference occurs in the middle. Corresponding results for $N = 3$ and 4 are shown in Supplementary Figs. 2 and 3.

The behavior would clearly be equivalent if, for a fixed sample with $L/a_z = 40$ and $N = 4$, we had applied the force at unit cell at $z/a_z = 39$ instead of $z/a_z = 40$. This means that the observed behavior is a violation of Saint Venant’s principle [13], which essentially states that moving the point at which a force ...”

“... The displacement $u_z(z)$ shown in Fig. 5d is distinctly different from the behavior right next to it on the left in that no oscillations occur at that end of the beam at which the force is applied to two unit cells at once. Here, the boundary conditions simply lead to a negligibly small amplitude of the frozen evanescent phonon mode decaying away from that end, as can be seen from the decomposed displacement field shown in Fig. 6c.

The described dependencies on the integers L/a_z and N result from different forces acting within the sample. One would therefore expect, for example, that the effective Hooke's spring constant of the beam ...”

The new figure 6 and its caption are:

“ ...

Figure 6. Decomposition of displacement field. (a) Result for the case of $N = 2$ and $L/a_z = 39$ in Fig. 5c. The total displacement (cf. red dots) is the solution of the mass-and-spring model shown in Fig. 5b. It is composed of a non-Bloch part $u_z^{\text{NB}} = c_1 + c_2 z$ (cf. gray dots), a frozen evanescent phonon eigensolution localized to the left end (cf. green dots), and a frozen evanescent phonon eigensolution localized to the left end (cf. purple dots) of the finite-length beam, respectively. See Methods for details on the decomposition. The two frozen evanescent phonon eigensolutions exhibit in-phase displacement fields in the middle of the beam, leading to constructive interference in the middle (cf. red dots). The straight lines connecting the dots are merely a guide to the eye. (b) Same as (a) but for $N = 2$, $L/a_z = 40$. Here, the total displacement (cf. red dots) in the middle shows a smaller oscillation amplitude than in (a) due to destructive interference of the two frozen evanescent phonon eigensolutions. (c) Same as (a), but for $N = 2$, $L/a_z = 40$ under the two-sites loading condition (cf. Fig. 5d). This boundary condition significantly suppresses the frozen evanescent phonon to the right end of the beam. See Supplementary Figs. 2 and 3 for other examples ($N = 3, 4$) corresponding to Fig. 5c.”

The new Supplementary Figs. 2 and 3, and their captions are:

“ ...

Supplementary Figure 2. Same as Fig. 6a but for (a) $N = 3, L/a_z = 38$, (b) $N = 3, L/a_z = 39$, and (c) $N = 3, L/a_z = 40$.

Supplementary Figure 3. Same as Fig. 6a, but for (a) $N = 4, L/a_z = 37$, (b) $N = 4, L/a_z = 38$, (c) $N = 4, L/a_z = 39$, and (d) $N = 4, L/a_z = 40$. We note that there are two instead of one frozen evanescent phonon eigensolutions localized at the two ends of the finite-length beam. One has an

oscillation period of $2a_z$ (cf. green and purple dots), while the other oscillates with a period of $4a_z$ (cf. light green and light purple dots), consistent with the complex wavenumbers for the frozen evanescent phonons (cf. Methods in the main paper).”

We have re-numbered the other Supplementary figures accordingly.

The new section explaining the displacement field decomposition in the Methods is:

“ ...

Decomposition of displacement field

For the simplified finite-length mass-and-spring model in Fig. 5b, we denote the axial (z -direction) displacement of each mass by u_n with $n = 0, 1, 2 \dots, L/a_z$. We first focus on the masses in the bulk, i.e., $n = N, N + 1, \dots, L/a_z - N$. Each of these masses is connected to two immediate neighbors and two N -th nearest neighbors on both sides. Thus, the force-balance equation for the mass n reads:

$$K_1(u_{n+1} + u_{n-1} - 2u_n) + K_N(u_{n+N} + u_{n-N} - 2u_n) = 0, \quad n = N, N + 1, \dots, L/a_z - N. \quad (11)$$

Apparently, this set of equations always supports the linear non-Bloch solution

$$u_n^{\text{NB}} = C_1 + C_2 n, \quad n = 0, 1, 2 \dots, L/a_z, \quad (12)$$

C_1 and C_2 are constants. Additionally, multiple frozen evanescent phonons of the following form are possible

$$u_n^j = \exp\left(ik_j(na - L/2)\right), \quad n = 0, 1, 2 \dots, L/a_z. \quad (13)$$

Here, the complex wavenumber k_j satisfies the following condition

$$K_1 \sin^2\left(\frac{k_j a}{2}\right) + K_N \sin^2\left(N \frac{k_j a}{2}\right) = 0. \quad (14)$$

Equation (14) has infinitely many solutions for k_j . Here, we only need to consider k_j with its real part inside of the first Brillouin zone, $-\pi/a_z < \text{Re}(k_j) \leq \pi/a_z$, as other solutions are simply shifted by an integer multiple of $2\pi/a_z$, which does not influence the displacement solution Eq. (13).

For the considered parameter settings in Fig. 5c, we have the following solutions for k_j :

$$N = 2, \quad k_1 \approx (1 + 0.0364 i)\pi/a_z, \quad k_2 \approx (1 - 0.0364 i)\pi/a_z \quad (15)$$

$$N = 3, \quad k_1 \approx (0.6662 + 0.0217i)\pi/a_z, \quad k_2 \approx (-0.6662 + 0.0217i)\pi/a_z, \\ k_3 = -k_1, \quad k_4 = -k_2. \quad (16)$$

$$N = 4, \quad k_1 \approx (0.4998 + 0.0116i)\pi/a_z, \quad k_2 \approx (-0.4998 + 0.0116i)\pi/a_z, \\ k_3 = -k_1, \quad k_4 = -k_2, \\ k_5 \approx (1 + 0.0164i)\pi/a_z, \quad k_6 \approx (1 - 0.0164i)\pi/a_z. \quad (17)$$

The displacement field in Eq. (13) for k_j with a positive imaginary part represents a frozen evanescent phonon that exponentially decays to the right, while the one with negative imaginary part stands for a frozen evanescent phonon exponentially decaying to the left.

The general displacement field of the finite-length system in Fig. 5b is a linear combination of the above solutions, i.e.,

$$u_n = C_1 + C_2 n + \sum_j E_j \exp(ik_j n a). \quad (18)$$

The displacement field Eq. (18) automatically ensures the force-balance equation Eq. (11) for the masses in the bulk, i.e., $n = N, N + 1, \dots, L/a_z - N$. The unknown complex coefficients, C_1, C_2 , and E_j , can be determined from the boundary conditions applied to the mass-and-spring chain.

For the single-site loading as in Fig. 5c, the left most mass and the right most mass have prescribed displacements, $u_0 = 0$, and $u_{L/a_z} = u_{\max}$. In addition, masses $n = 1, \dots, N - 1$, and masses $n = L/a_z - N + 1, \dots, L/a_z - 1$ must be in force-balance, too. For the two-sites loading condition in Fig. 5d, the mass $n = L/a_z - 1$ is similarly prescribed with displacement, $u_{L/a_z - 1} = u_{\max}$. Other conditions are the same as the single-site loading conditions. It can be easily verified that the number of the boundary conditions, $2N$, exactly matches the number of the unknown coefficients. Thus, the unknown coefficients can be determined. Afterwards, the total displacement field Eq. (18) as well as individual parts, including the linear part and the frozen evanescent phonons, can be obtained. The analytical formula for the unknown coefficients is very lengthy and thus omitted here.

..."

Finally, we have changed the main text on page 6 as follows:

"...This behavior results from the fact that the static solution is a linear superposition of all frozen-phonon Bloch eigenmodes and of the **general** non-Bloch solution for a finite-size sample $u_z^{\text{NB}}(z) = c_1 + c_2 z$, with constants c_1 and c_2 . The latter is always a solution because the underlying equation of motion contains only up to the **second-order spatial derivative**. The oscillations of $u_z(z)$ versus z lead to regions of negative slope. This means that the structure is locally compressed in these regions – despite the fact that the structure is elongated overall.

..."

The rest of the arguments of the paper are well justified in my opinion. With the addition suggested above, I suggest publication of this article in Nature Communications.

To our understanding, we have fully included the additions suggested by the reviewer #1. On this basis, reviewer #1 suggests publication of this article in Nature Communications. We once again thank reviewer #1 for the constructive suggestions.

Reviewer #2

This manuscript introduces frozen evanescent phonons as eigenstates of the phonon band structure characterized by complex-valued wavenumbers that freeze at zero frequency. It commences with a thorough discussion of the analytical nature of the phonon band structure, elucidating how extrema in the band structure correspond to the presence of evanescent modes. The study further investigates the influence of boundary conditions and specimen size on the response of finite-size materials to external forces, uncovering notable violations of Saint Venant's principle. Additionally, the manuscript provides mechanism-based metamaterials as an alternative approach for generating local minima in the supplemental materials and also offers insights into the two-dimensional case. The findings of this

manuscript are intriguing, and the comprehensive evidence provided through experimental simulations and theoretical analyses adds substantial weight to the conclusions. I would recommend potential publication once the addressed concerns have been appropriately resolved.

We thank reviewer #2 for their detailed and very positive assessment of our manuscript.

1. At the extrema of the band ω_i , the Taylor expansion ignores the 1st derivative term, is it because near the extrema the quadratic characteristic dominates? In line 72, the author calculates ζ from the 2nd derivative term. Should there be a 1/2 factor included? as the coefficient of the 2nd derivative term should be $f''/2$.

We apply the Taylor expansion to an extremum on a dispersion band $\omega_i(k)$, either a local maximum or a local minimum. For any local maximum or local minimum, the 1st derivative is strictly zero – by definition. We do not feel that we should explain this basic fact in the paper.

We thank the reviewer for the careful reading and for noticing the error in the calculation of ζ . The error actually comes from forgetting to type the factor of 1/2 in the Taylor expansion of the dispersion relation. We have fixed the mistake in the revised version of the paper. With this additional factor of 1/2, the calculated ζ and the decay length l in the original version of our paper are both correct. The revision in the main paper is:

“ ...

Let us start our discussion from general principles, independent from any particular experimental realization: The phonon band structure $\omega_i(k)$ results from a physical equation of motion of the system and is thus an analytical function, at least in the vicinity of the real axis in the complex plane [15, 16]. Consider a minimum or maximum of the real frequency ω_i in the band structure of band i at real wavenumber $k_{\min/\max}$ illustrated in Fig. 1. In the vicinity of such an extremum, we can Taylor expand

$$\omega_i(\text{Re}(k)) = \omega_{\min/\max} + \frac{1}{2}\zeta(\text{Re}(k) - k_{\min/\max})^2 + \dots$$

Mathematically, frequency ω_i and wavenumber $k = \text{Re}(k) + i \text{Im}(k)$ can be complex numbers.

...”

2. I don't see the finite element data in Figure 5. Were they forgotten to be added or are they being covered by the experimental and theoretical data?

The agreement between the finite-element data and the other data is so good that the symbols largely overlap. To improve the readability of Fig. 5, we have slightly enlarged the markers (blue dots) for the finite-element data and also revised the caption as below:

“ ...

Figure 5. Anomalous displacement fields of stretched metamaterial beams. (a) Side-view optical micrograph of a metamaterial beam sample (also see Fig. 2 and Fig. 4), extending from $z = 0$ (glass-substrate side) to $z = L$. (b) Simplified representation in terms of a mass-and-spring model. (c) A force along the z -direction is exerted at the last unit cell of the samples at $z = L$, leading to a stretching of the samples along the z -direction by an engineering strain of $u_{\max}/L \approx 1\%$. The resulting displacement-vector component u_z is recorded as a function of the site number or relative position z/a_z . Orange dots refer to experimental measurements, blue dots to finite-element numerical calculations, and red dots to solutions of the mass-and-spring model. **We note that the finite-element results largely overlap with the other data within the symbol size, indicating excellent agreement.** The integer parameters N and L_z/a are indicated in the subpanels. For an ordinary elastic material, the displacement field..."

3. Is it possible to have multiple local minima in the 1D system? If so, what will the response look like when applying quasi-static loading?

Yes, multiple local minima of the dispersion relation can be achieved in the one-dimensional system for the case of higher-order nonlocal interactions. We refer the reviewer to Supplementary Fig. 1b,

where we show the calculated complex phonon band structures for a nonlocal metamaterial beam with 4th-order nonlocal interactions ($N = 4$):

“ ...
a

b

Supplementary Figure 1. Same as Fig. 3a but for metamaterials with (a) $N = 2$ and (b) $N = 4$. For $N = 2$, the longitudinal branch (blue curve in (a)) exhibits a local minimum at the edge of the first Brillouin zone, leading to a frozen evanescent mode with wavenumber $k_z = (1 - 0.044 i)\pi/a_z$ (cf. blue dot in (a)). In contrast, the longitudinal branch for $N = 4$ (blue curve in (b)) shows two local minima and two resulting frozen evanescent modes at wavenumbers, $k_z = (0.5 - 0.013 i)\pi/a_z$ and $k_z = (1 - 0.033 i)\pi/a_z$ (cf. two blue dots in (b)).

...”

In order to make this point more explicit to the reader, we have slightly revised the text connecting to Fig. 3 in the main paper:

“ ...

Results similar to the ones shown in Fig. 3 for $N = 3$, but for $N = 2$ and $N = 4$, are depicted in Supplementary Fig. 1. **The number of local minima increases with increasing N . Likewise, more frozen evanescent phonons arise.** More broadly, many different metamaterial approaches can lead to local minima in the dispersion relation, likewise leading to anomalous frozen evanescent phonon Bloch modes. This includes mechanism-based metamaterials [40] and others [41]. However, the corresponding effects become only pronounced for very low-frequency local minima, leading to large characteristic exponential decay lengths.

...”

The static response of a metamaterial beam with multiple local minima on its dispersion band does not exhibit a conceptual difference compared to metamaterials with a single local minimum. The displacement field generally exhibits spatial oscillations due to excited frozen evanescent phonons. The reviewer can compare the results with $N = 2$ and $N = 4$ in Fig. 5c. The difference becomes more pronounced when checking the decomposed displacement field, including a linear part and frozen evanescent phonon parts, as suggested by reviewer #1 (see above). We observe multiple frozen evanescent phonons localized at the same end in contrast to a single frozen evanescent phonon for metamaterials with a single local minimum on its dispersion band. We also refer the reviewer to our

response to the second comment of reviewer #1. Reviewer #2 can also compare the newly added Supplementary Figs. 2 and 3.

We feel that these aspects are sufficiently discussed in the revised version of our manuscript due to the changes made in response to reviewer #1.

4. In Supplementary Fig. 3, the authors illustrate anomalous frozen evanescent modes in mechanism-based mechanical metamaterials. It appears that the mode is the floppy mode. Does this imply that any mechanical metamaterials with floppy modes have the potential to realize this effect? Could the authors provide insights into the connection between the floppy mode and frozen evanescent phonons? Additionally, it seems that the red dot is missing in panel b.

The frozen evanescent phonon of the mechanism-based metamaterial is not identical to a floppy mode of the metamaterial, but it is related. In fact, when the hinges connecting the squares in the metamaterial become ideal point-like connections, the local minimum of the blue band (cf. Supplementary Fig. 5 in the revised version, Supplementary Fig. 3 in the original version) at the edge of the Brillouin zone will eventually touch zero frequency. The corresponding Bloch mode then becomes a perfect floppy mode, i.e., a deformation mode that costs zero strain energy. For any finite-size hinges, the zero eigenfrequency is lifted to a finite-valued local minimum of the band, leading to frozen phonons by the Cauchy-Riemann principle.

In general, a metamaterial with floppy modes, i.e., non-trivial Bloch modes with zero eigenfrequency is a good starting point to realize frozen evanescent phonons. As implied above, a small parameter perturbation can be introduced into the metamaterial to achieve a local minimum of the dispersion band and thus to realize frozen evanescent phonons. However, not all frozen evanescent phonons are related to floppy modes, such as frozen evanescent phonons that originate from local minima of higher bands with frequencies far above zero frequency.

We address these points in the revised version of the main paper:

“... In Supplementary Fig. 5, we discuss a mechanism-based metamaterial [40] as another example. A local minimum of the dispersion band occurs at the edge of the first Brillouin zone. This local minimum and the resulting frozen evanescent phonons are connected to a floppy mode of the mechanism-based metamaterial with ideal-hinge connections. The authors [40] described a route towards obtaining large characteristic lengths in mechanics that did not involve band structures or (evanescent) Bloch modes. Our approach shades a different light onto these periodic metamaterials rather based on their Bloch eigenstates. ...”

We have also revised the caption of Supplementary Fig. 5 (Supplementary Fig. 3 in the original version):

Supplementary Figure 5. Another example of anomalous frozen evanescent modes in mechanism-based mechanical metamaterial. (a) Metamaterial [1] consisting of squares (light yellow) connected by non-ideal hinges (cf. small blue squares). The colors are for illustration only; all is made from a single constituent material. The hinges become ideal if the blue squares become infinitely small, leading to a floppy mode (with $\text{Im}(k_z) = 0$ at $\text{Re}(k_z) = \pi/a_z$) or mechanism of deformation (neighboring squares rotate oppositely) at zero energy cost. (b) Calculated complex-valued phonon band structures of the metamaterial. Here, we choose the same constituent-material parameters as in Fig. 3 and the geometry parameters $d/a_z = 0.05$ and $a_z = 1$ cm. The wavevector, $\mathbf{k} = (0, 0, k_z)$, is along the beam axis. Frozen evanescent phonon modes ($k_z = (1 + 0.09 i) \pi/a_z$) (cf. blue dot in (b)) emerge from the low-frequency local minimum of the blue band at the edge of the first Brillouin zone. The local minimum further approaches zero frequency for yet smaller ratios d/a_z (not depicted). (c) Displacement field of the frozen evanescent mode (cf. blue dot in (b)). Neighboring squares along the z -direction rotate in opposite directions, consistent with the period of $p = 2\pi/\text{Re}(k_z) = 2a_z$.

...”

We thank the reviewer for noticing the missing “red” dot. We actually meant the “blue” dot in the caption of Supplementary Fig. 5 (Supplementary Fig. 3 in the original version). This mistake has been fixed in the revised version as indicated above.

5. The authors present the eigenmodes for the nonlocal model in Supplementary Fig. 5, which display pronounced oscillations with a spatial period of $3a$. To observe these frozen evanescent modes, what type of excitation should be applied, and where should it be applied? Should it be a quasi-static loading similar to the 1D case, or should it be a load with the real part of the eigenfrequency?

To observe the shown finite-frequency eigenmodes, which must not be confused with the frozen evanescent phonon modes, in Supplementary Fig. 7d (Supplementary Fig. 5d in the original version), the excitation frequency must be close to one of the eigenfrequencies. The mode of excitation is not critical. To illustrate this statement, we have added the new Supplementary Fig. 8:

“

Supplementary Figure 8. Static and dynamic response of the 2D nonlocal system in Supplementary Fig. 6. (a) Displacement field of the system under static loading at the center of the membrane. The four edges of the system are fixed. (b)-(e) Same as (a), but for harmonic time-harmonic excitation at a frequency close to one of the first four eigenfrequencies, i.e., $0.99\omega_1$, $0.99\omega_2$, $0.99\omega_3$, $0.99\omega_4$, respectively (in the absence of damping, exciting the system exactly at an eigenfrequency leads to a divergence). In both static case and dynamic case, spatial oscillations with period $3a$ are observed – as for the eigenmodes shown in Supplementary Fig. 7d.”

In fact, for the case of static loading, one also observes that the displacement field has a spatial oscillation period of $3a$, corresponding to the “wavelength” of the frozen evanescent phonons in the two-dimensional system.

This new figure is referred to in the main paper on page 8:

“ ...

Our discussion so far has been restricted to effectively one-dimensional decay problems for which one local minimum in the dispersion relation $\omega_i(k)$ leads to four frozen evanescent phonon modes per branch i (see Fig. 1b and Fig. 3a). In effectively two-dimensional problems (see Supplementary Fig. 6), we find infinitely many frozen evanescent eigensolutions at $\omega = 0$, forming an area within the plane spanned by $\text{Re}(k_x)$ and $\text{Re}(k_y)$ (see Supplementary Fig. 7). For example, when exerting a point-like force onto a two-dimensional thin square-shaped membrane fixed at all of its four edges, the membrane’s static out-of-plane displacement spatially oscillates within the plane (cf. Supplementary Fig. 8). These pronounced spatial oscillations also occur in the lowest-frequency eigenmodes of the finite membrane (see Supplementary Figs. 6 and 7), including the fundamental mode, which usually exhibits only a single maximum in the middle of the membrane. Here, we move upwards on the evanescent branch starting from the black dot at $\omega = 0$ in Fig. 1b. The modes can be excited by time-harmonic excitation at frequencies close to the membrane eigenfrequencies (cf. Supplementary Fig. 8).

...”

REVIEWERS' COMMENTS

Reviewer #1 (Remarks to the Author):

I am fully satisfied with the revised version of the paper, hence recommend publication in Nature Communications.

One minor point that I would like to bring to the authors' attention: The authors claim that Jackiw-Rebbi method has only been applied to finite frequency evanescent modes before. This is not quite true. It has also been applied to understand static (frozen) domain wall and corner modes in mechanical metamaterials (see, for example, Phys. Rev. B 108, L060103 (2023)).

Reviewer #2 (Remarks to the Author):

The changes the authors made have improved the manuscript and the response has addressed my previous concerns. I would like to provide my recommendation for its publication.

Response to Reviewers

We thank the reviewers for their response. In what follows, *we repeat their comments in red and italics*, we respond in green, we cite from the original manuscript in black, and highlight changes made to the manuscript in blue.

Reviewer #1

I am fully satisfied with the revised version of the paper, hence recommend publication in Nature Communications.

We thank the reviewer for recommending publication of our paper.

One minor point that I would like to bring to the authors' attention: The authors claim that Jackiw-Rebbi method has only been applied to finite frequency evanescent modes before. This is not quite true. It has also been applied to understand static (frozen) domain wall and corner modes in mechanical metamaterials (see, for example, Phys. Rev. B 108, L060103 (2023)).

We have cited the reference (Ref. [10]) pointed out by the reviewer in our Introduction:

“...

Ordinarily, evanescent waves are associated with *finite* frequencies inside frequency bandgaps [6]. A well-known example is the Jackiw-Rebbi solution [7-9] localized at a domain-wall separating two different topological phases. *Jackiw-Rebbi states have been applied to study static domain wall states and corner modes in mechanical metamaterials [10].* Here, we apply the concept of evanescent waves in single-domain samples to the *static* regime, in which the wave gets “frozen” in time. These special Bloch modes allow for making a direct connection between unusual static and dynamic properties, which has previously been unclear, hampering the rational design of unusual static behavior in metamaterials... “

We have re-numbered all references accordingly.

Reviewer #2

The changes the authors made have improved the manuscript and the response has addressed my previous concerns. I would like to provide my recommendation for its publication.

We thank the reviewer for recommending publication of our paper.